# Daily changes in phytoplankton lipidomes reveal mechanisms of energy storage in the open ocean

Kevin W. Becker [1], James R. Collins [1,2,6], Bryndan P. Durham[3], Ryan D. Groussman[3], Angelicque E. White [4], Helen F. Fredricks[1], Justin E. Ossolinski[1], Daniel J. Repeta[1], Paul Carini [5,7], E. Virginia Armbrust[3] & Benjamin A.S. Van Mooy [1]

Sunlight is the dominant control on phytoplankton biosynthetic activity, and darkness deprives them of their primary external energy source. Changes in the biochemical composition of phytoplankton communities over diel light cycles and attendant consequences for carbon and energy flux in environments remain poorly elucidated. Here we use lipidomic data from the North Pacific subtropical gyre to show that biosynthesis of energy-rich triacylglycerols (TAGs) by eukaryotic nanophytoplankton during the day and their subsequent consumption at night drives a large and previously uncharacterized daily carbon cycle. Diel oscillations in TAG concentration comprise $23 \pm 11\%$ of primary production by eukaryotic nanophytoplankton representing a global flux of about 2.4 Pg C $yr^{-1}$. Metatranscriptomic analyses of genes required for TAG biosynthesis indicate that haptophytes and dinoflagellates are active members in TAG production. Estimates suggest that these organisms could contain as much as 40% more calories at sunset than at sunrise due to TAG production.

[1] Department of Marine Chemistry and Geochemistry, Woods Hole Oceanographic Institution, Woods Hole, MA 02543, USA. [2] Massachusetts Institute of Technology/Woods Hole Oceanographic Institution Joint Program in Oceanography, Woods Hole, MA 02543, USA. [3] School of Oceanography, University of Washington, Seattle, WA 98195, USA. [4] College of Earth, Ocean, and Atmospheric Sciences, Oregon State University, Corvallis, OR 97331, USA. [5] Department of Microbiology, Oregon State University, Corvallis, OR 97331, USA. [6]Present address: School of Oceanography and eScience Institute, University of Washington, Seattle, WA 98195, USA. [7]Present address: Department of Soil, Water and Environmental Science, University of Arizona, Tucson, AZ 85721, USA. Correspondence and requests for materials should be addressed to B.A.S.V.M. (email: bvanmooy@whoi.edu)

Diel cycles in sunlight represent an important source of environmental variability for plankton communities in the oligotrophic subtropical gyres of the world's ocean. This is particularly true for phytoplankton, which depend on sunlight to drive the biosynthesis of macromolecules essential for growth. These organisms possess numerous metabolic capabilities that allow them to survive in darkness, when they are effectively deprived of their primary energy source. For example, diel cycles in phytoplankton gene expression patterns reveal widespread coordination of metabolic pathways to optimize internal biochemical energy allocation[1–3]. However, the attendant diel cycles in the biochemical composition of phytoplankton communities and the biogeochemical consequences of these changes for the ocean carbon cycle remain poorly understood. As subtropical gyres of the oligotrophic ocean are the world's largest biomes, account for about 40% of the Earth's surface area, and contribute significantly to global productivity and carbon export[4,5], diel cycles in phytoplankton metabolism are expected to be engrained in the roles these environments play in the global carbon cycle.

A common strategy for energy allocation in phytoplankton centers on the daytime accumulation of energy storage molecules that can then be used to drive metabolism and respiratory activity during the night[6,7]. Laboratory cultures of phytoplankton display clear diel cycles in storage lipid accumulation[8–10], but whether this metabolic behavior is manifested in the ocean is an open question. While cyanobacteria store energy in the form of carbohydrates (mainly glycogen), eukaryotic phytoplankton predominantly accumulate neutral lipids, mainly in the form of triacylglycerols (TAGs). TAGs are more effective energy stores than carbohydrates, because they contain more chemical energy per mole of carbon and larger quantities can be stored inside the cell as they are non-polar and anhydrous (i.e., lacking a hydration shell)[11]. Thus, large phytoplankton, such as diatoms and dinoflagellates, have a greater capacity for energy storage when compared with small cyanobacterial phytoplankton, such as *Prochlorococcus* and *Synechococcus*[12]. The accumulation of energy storage molecules in the form of glycogen and/or TAGs is apparently also a strategy for phytoplankton to deal with unfavorable growth conditions[13]. In the oligotrophic surface waters of the subtropical gyres, nitrogen is persistently scarce; this scarcity has been linked in cultures of many eukaryotic phytoplankton to enhanced TAG accumulation[13]. Thus, in subtropical gyres, we hypothesize that TAGs could play a number of roles in eukaryotic phytoplankton metabolism, contribute to the ability of eukaryotic phytoplankton to compete with cyanobacteria, and thereby play a previously uncharacterized role in the carbon cycle of these environments.

Oscillations in phytoplankton storage lipids have the potential to play important roles in biogeochemical carbon cycling and energy flow through the broader microbial community. For example, the biochemical composition of phytoplankton defines their nutritional quality[14,15], which has implications for the population dynamics of the zooplankton that graze on them. Yet, field studies on diel periodicity of phytoplankton biochemical composition are rare[16–18]. To this end, we investigated the lipid composition of the planktonic community (i.e., the metalipidome) at high temporal resolution over multiple day/night cycles during two research cruises to the North Pacific Subtropical Gyre (NPSG). Complementary analyses of the plankton community metatranscriptome were conducted to help constrain the sources of the lipids we observed. Comparison to measurements of primary productivity showed that diel synthesis and consumption of TAGs account for a large fraction of the flux of carbon and energy through the ecosystem.

## Results

**Diel patterns of phytoplankton lipids.** During a cruise in the NPSG to Station ALOHA (22°45'N, 158°W) in summer of 2015, we collected samples within sunlit surface waters (15 m) every 4 h over the course of 8 days. We then conducted downstream lipidomic analyses to determine the concentrations of several major lipid classes (Fig. 1). A survey, conducted at a courser time resolution over 3 days in oligotrophic North Atlantic, suggested that small fluctuations in intact polar membrane lipids were possible in oligotrophic ocean environments[18]. The betaine lipid diacylglycerylcarboxy-N-hydroxymethyl-choline (DGCC) is an intact polar membrane lipid that is associated with eukaryotic phytoplankton in samples from the oligotrophic regions[19]. The time-of-day average (Fig. 1b) of DGCC was highest at dusk (~18:00 h local time) and lowest at dawn (~06:00 h), but the $1.27 \pm 0.2$-fold difference was not significant ($t$-test, $t(\mathrm{df} = 8) = -1.947$, $p = 0.0874$). Furthermore, diel (24-h) periodicity was not detected for concentrations of this lipid class ($p > 0.05$) using the RAIN (Rhythmicity Analysis Incorporating Nonparametric methods) algorithm[20]. DGCC is expected to scale nearly linearly with the number of cells[21], and thus the relatively stable concentrations of DGCC at Station ALOHA indicates that the number of cells was also similarly invariant throughout the diel cycle of sunlight, i.e. production of cells was balanced by grazing and other mortality factors.

In contrast, TAG concentrations showed significant diel periodicity (RAIN algorithm, $p = 0.004$) with dramatic oscillations over the diel cycle ($2.3 \pm 0.4$-fold mean ± SD; $t$-test, $t(\mathrm{df} = 8) = -6.200$; $p = 0.0002$). Highest abundances in TAGs occurred during the day near dusk, and lowest abundances at dawn. The significantly greater oscillations in TAG concentrations relative to DGCCs (and relative to particulate organic carbon (Supplementary Figure 1)) indicates that the TAG signal is not simply a reflection of cell size or abundance, but is instead consistent with the synthesis and accumulation of TAGs by cells during daylight hours and depletion at night. Eukaryotic phytoplankton have been proposed as the main source of TAGs in the ocean[22] and the distribution of individual molecular species of TAGs is consistent with other reports from eukaryotic phytoplankton from the Pacific with a dominance of TAGs containing 48, 50, and 52 carbon atoms and one unsaturation[22] (Supplementary Figure 2). To confirm that eukaryotic phytoplankton dominate the TAG signal in the NPSG, we screened several strains of *Prochlorococcus* and *Pelagibacter*, the numerically dominant genera of cyanobacteria and heterotrophic bacteria, respectively, in oligotrophic gyres[23,24]. These two groups account for more than half of the biomass in these regions[25]. No TAGs were detected in any of the strains, consistent with the lack of TAG biosynthesis genes in both groups. Other bacteria (e.g., from the genera *Mycobacterium*, *Streptomyces*, and *Rhodococcus*) known to produce TAGs[25] are not abundant in surface waters of the NPSG[26]. By contrast, the eukaryotic phytoplankton in the oligotrophic gyres are dominated by haptophytes, dinoflagellates, diatoms, and prasinophytes[27,28], and all of these groups have the ability to synthesize TAGs[13]. The relative magnitude of TAG increase we observed is in the range of what has been observed in culture experiments. In the haptophyte *Isochrysis* sp. grown under N-limitation, neutral lipid concentrations, which were likely mostly TAGs, increased by ~1.7-fold between the beginning and end of the photoperiod[29]. Similarly, in the dinoflagellate coral symbiont *Symbiodinium*, the quantity of TAGs in lipid bodies increase by approximately twofold[17]. Thus, we concluded that the marked diel cycles in TAG concentrations we observed in the NPSG were most likely driven by eukaryotic phytoplankton.

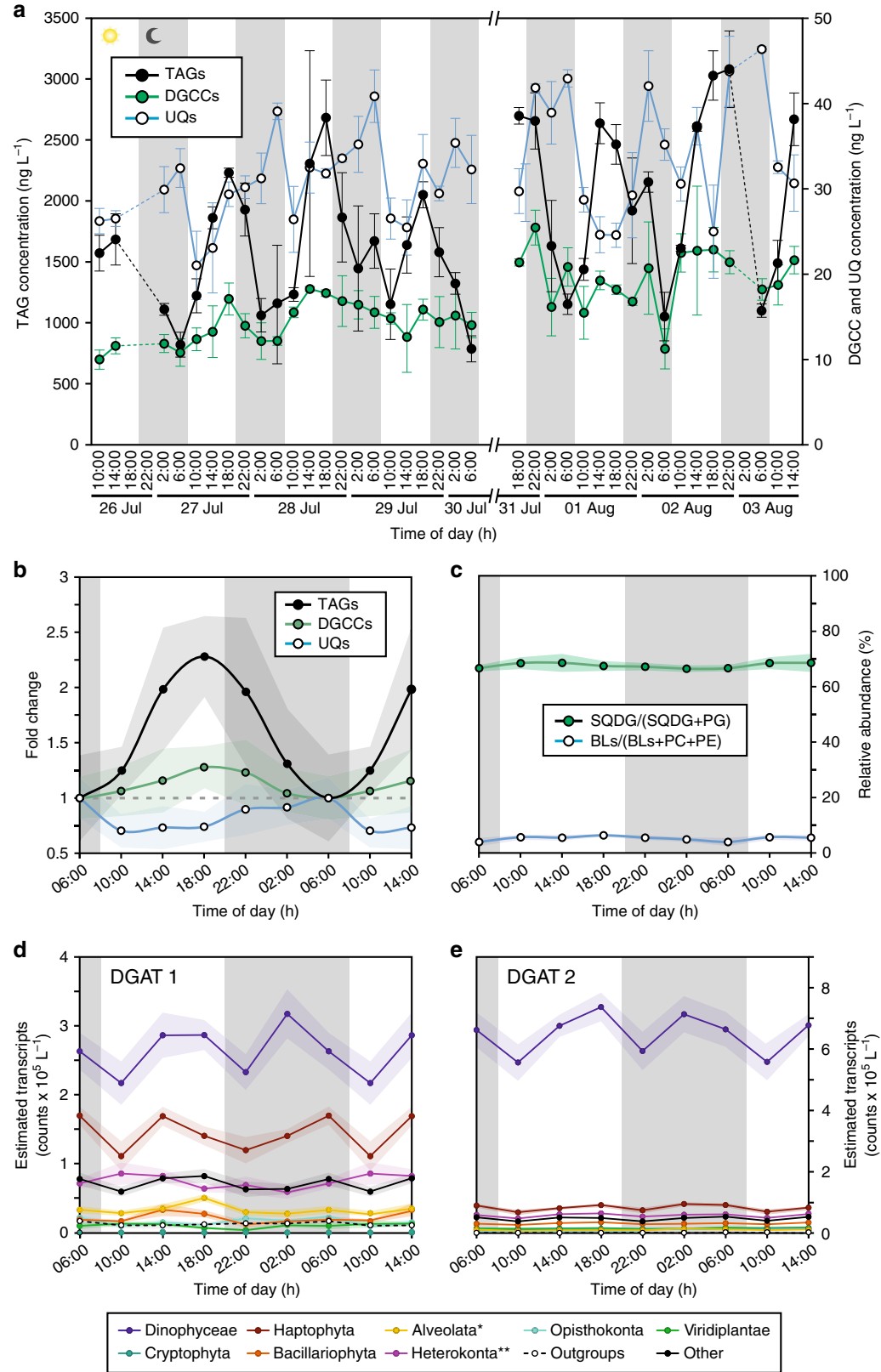

Another class of lipids, the ubiquinones (UQs), also displayed a diel periodicity in concentration (RAIN algorithm, $p = 0.002$), but the phase was opposite of TAGs with a pre-dawn maximum followed by a precipitous drop to an early-morning minimum ($0.3 \pm 0.1$-fold mean $\pm$ SD; $t$-test, $t(\mathrm{df} = 8) = 3.616$, $p = 0.0068$) (Fig. 1a and b). UQs are essential components of the electron transport chain used for cellular respiration by most aerobic organisms[30]. In the NPSG, these lipids are likely derived primarily from eukaryotic microbes as cyanobacteria do not synthesize UQs – cyanobacteria use plastoquinones for photosynthetic electron transport and respiration[31]. Furthermore, we observed that the distribution of different UQ molecules in the

**Fig. 1** Diel oscillation of lipid biomarkers. **a** Concentrations of storage lipids (TAGs), a class of structural membrane lipids from eukaryotic phytoplankton (DGCCs) and lipid-soluble electron transporters (UQs) over 8 day/night cycles at 15 m water depth in July and August 2015 at Station ALOHA. Values represent the average of environmental triplicates and error bars the standard deviation. **b** Time-of-day average of fold change relative to 06:00 h (local time) of TAG, DGCC and UQ concentrations. **c** Ratios of intact non-phosphorus membrane lipids (SQDG and BLs) to membrane phospholipids (PG, PE, and PC). Lipid ratios are based on known lipid substitutions in eukaryotic phytoplankton and cyanobacteria under nutrient limitation[42]. **d** and **e** Time-of-day average of diacylglycerol acyltransferase 1 (DGAT1) and DGAT2 abundance, respectively, in major eukaryotic phytoplankton (* not including Dinophyceae (dinoflagellates), ** not including Bacillariophyta (diatoms)). Shaded areas around lines in **b** and **c** represent the standard deviation of averaged values for each time of day ($n \geq 6$). Vertical gray shaded areas in each graph indicate night. TAG triacylglycerol; DGCC diacylglycerylcarboxy-*N*-hydroxymethyl-choline, UQ ubiquinone; SQDG sulfoquinovosyl diacylglycerol; PG phosphatidylglycerol; BL betaine lipid (sum of DGCC, diacylglyceryl trimethylhomoserine (DGTS), and diacylglyceryl hydroxymethyl-trimethyl-β-alanine (DGTA)); PC phosphatidylcholine; PE phosphatidylethanolamine

NPSG differs from that of *Pelagibacter* (Supplementary Figure 3). Respiratory quinone concentrations in cultures and environmental samples have been suggested to reflect respiratory activity of their producers[32–34] and the observed diel oscillations of UQs may indicate enhanced cellular respiration at night, which is consistent with the consumption of TAGs by phytoplankton. In addition, it has recently been proposed, based on genomic analysis, that increased ubiquinone levels might be required in *Escherichia coli* cells when grown on long-chain fatty acids[35]. Ubiquinones exhibit antioxidant properties and could mitigate elevated levels of reactive oxygen species generated by fatty acid degradation. That this strategy occurs in eukaryotic phytoplankton remains to be shown, but it could explain our observation of higher ubiquinone concentrations at night when TAGs are degraded.

**Metatranscriptome analysis of the plankton community.** Within the NPSG, the phytoplankton community is dominated by cyanobacterial and eukaryotic picophytoplankton cells less than ~2–3 μm and eukaryotic nanophytoplankton in the 2–20 μm size range[36]. To identify which eukaryotic phytoplankton were the likely sources of TAGs in the NPSG, we analyzed metatranscriptome expression patterns of two genes that encode characterized diacylglycerol acyltransferase isoforms (DGAT1 and DGAT2), both of which commit the final step in TAG biosynthesis[37]. Thus, a transcriptional signal of these genes is expected for all cells producing TAGs regardless of how the observed diel cycle is regulated. In most eukaryotic algae, DGAT1 is encoded by a single-copy gene, whereas DGAT2 is often encoded by multiple gene copies[38]. *DGAT1* is widely distributed across marine phytoplankton, with distinct clades of sequences that largely reflect taxonomic groupings. Notably, however, cultured isolates of the smallest and most abundant eukaryotic picophytoplankton, the prasinophytes *Ostreococcus* and *Micromonas*, lack *DGAT1* and instead rely on multi-copy *DGAT2* for TAG biosynthesis[38]. In the NPSG, haptophyte and dinoflagellate taxa dominate the *DGAT1* transcriptome signal (~$6.4 \times 10^5$ counts L$^{-1}$), together accounting for almost 70% of the detected *DGAT1* transcripts (Table 1; Supplementary Figure 4), regardless of time of day. Despite the distinct diel pattern in TAG abundance, transcript abundance for the *DGAT1* gene remained relatively constant throughout the four days of transcript sampling (Fig. 1d), suggesting that TAG abundance in the NPSG might be regulated through intracellular consumption instead of production. Dinoflagellates, which accounted for the majority of *DGAT1* transcripts, are known for their lack of transcriptional regulation[39] and an absence of transcriptional rhythmicity within the Haptophyta clade was also expected given that in culture, the haptophyte *Chrysochromulina tobin*, displays no diel oscillations in *DGAT1*[40]. We also examined transcript abundance associated with *DGAT2* (Fig. 1e) to determine the role of prasinophytes in TAG biosynthesis. As with the *DGAT1* gene, transcript

**Table 1 Relative abundance of *DGAT1* transcripts in the NPSG**

| Taxonomic group[a] | Relative abundance (%)[b] |
|---|---|
| Opisthokonts, glaucophytes, cryptophytes | 6.4 |
| *Oikopleura dioica* | 1.1 |
| *Lepeophtheirus salmonis* | 2.3 |
| Alveolates[c] | 4.4 |
| *Strombidium inclinatum* | 3.5 |
| Dinophyceae (dinoflagellates) | 42.5 |
| *Ceratium fusus* | 8.8 |
| *Gymnodinium catenatum* | 9.6 |
| *Noctiluca scintillans* | 2.4 |
| *Scrippsiella trochoidea* | 1.4 |
| *Bolidomonas sp* | 4.4 |
| Heterokonts[d] | 12.1 |
| Viridiplantae (green algae) | 1.4 |
| *Paraphysomonas imperforata* | 1.0 |
| Haptophytes | 25.3 |
| *Chrysochromulina polylepis* | 1.3 |
| *Chrysochromulina brevifilum* | 11.8 |
| Bacillariophyta (diatoms) | 3.2 |

[a]Taxonomic groups represent the best read placements in the maximum-likelihood phylogenetic tree constructed from publicly available marine genomes and transcriptomes (see Supplementary Figure 4)
[b]Relative abundances for species within clades are given when recruited reads were >1% of total reads
[c]Not including Dinophyceae
[d]Not including Bacillariophyta

abundance (~$9.1 \times 10^6$ counts L$^{-1}$) for the *DGAT2* gene across the different clades remained constant over the 4 days, although potential diel cycles for specific copies could be obscured by this analysis; culture studies with *C. tobin* indicate that only one of the nine *DGAT2* gene copies displays a diel cycle in transcript abundance[40]. The *DGAT2* transcript signal was also dominated by dinoflagellate taxa with few (average <1%) transcripts detected for prasinophytes (Fig. 1e). A number of genes involved in the production of the precursors to TAGs, phosphatidic acids (e.g., acyltransferases) and in the consumption of TAGs (e.g., lipases) were also examined, and none showed regular diel variations, with the exception of a single, relatively scarce acyltransferase and a single acyl-CoA binding protein from haptophytes (Supplementary Table 1). Together the metatranscriptome data suggest that eukaryotic nanophytoplankton, particularly the haptophytes and dinoflagellates, dominate TAG production in the NPSG.

**TAG production rates throughout the photic zone.** In culture experiments, nutrient limitation can be the major stimulus for TAG accumulation in phytoplankton[13] and the growth rate of eukaryotic phytoplankton in the NPSG has been shown to be limited by nutrients[40,41]. An archetypal microbial response to nutrient stress, particularly phosphorus stress, is the substitution of non-phosphorus membrane lipids for phospholipids; this

response has been observed in nearly all of the major classes of plankton in the NPSG, including *Prochlorococcus*, *Synechococcus*, *Pelagibacter*, diatoms, heterokonts, and haptophytes[21,42–44]. Furthermore, we have previously shown that differences in the ratios of non-phosphorus lipids to phospholipids between the oligotrophic ocean ecosystems, including the NPSG, reflect differences in ambient phosphate concentrations and turnover rates[42]. These ratios of membrane lipids showed no significant diel variations over the investigated time interval (Fig. 1c), suggesting the absence of diel fluctuations in nutrient stress. In haptophytes, TAGs, along with diagnostic viral glycosphingolipids, have also been shown to accumulate in response to viral infection[45], but these viral glycosphingolipids were not detected in any of the NPSG samples we analyzed. Thus, the diel TAG cycle in the NPSG was not likely a reflection of daily fluctuations in nutrient stress or viral mortality.

We posit instead that the magnitude of the diel signal in biosynthesis of TAGs is driven directly by sunlight availability. This is evidenced by depth profiles of net TAG concentrations collected at dusk and dawn in March 2016 (Fig. 2a), showing that the magnitude of diel periodicity observed in surface waters is dampened with depth and disappears below 100 m, the depth where net photosynthetic carbon assimilation approaches zero (Fig. 2b). Accordingly, daily TAG production rates derived from

these concentrations were highest in the surface waters ($0.42 \pm 0.15$ mg TAG C $m^{-3}$ $d^{-1}$; $n = 6$) and decreased with water depth (Fig. 2b). The nitricline at Station ALOHA is typically at around 125 m[46], and thus decreasing nutrient stress with depth cannot explain the observed decrease in TAG production rate. The invariant molecular distribution of TAGs across different water depths (Fig. 2c) further suggests a similar biological source of TAGs throughout the euphotic zone, which is consistent with the major nanophytoplankton groups, diatoms, haptophytes, and dinoflagellates, being found throughout these waters in similar abundances[27]. Accordingly, the molecular distribution of DGCCs was highly similar at different water depths and different times of the day (Supplementary Figures 2 and 5).

**TAG contribution to primary production**. To estimate the contribution of TAGs to daily biomass production, we calculated net TAG production rates based on the difference in TAG concentrations between dawn and dusk, and compared these rates with net primary production rates determined during the same time interval (Table 2). In cells of some eukaryotic phytoplankton in culture, including dinoflagellates and haptophytes, TAGs can account for up to 50% of the cellular dry weight[13], illustrating the potential for a large investment of photosynthetic energy into TAG synthesis. The $^{14}$C-based primary productivity ($^{14}$C-PP) of

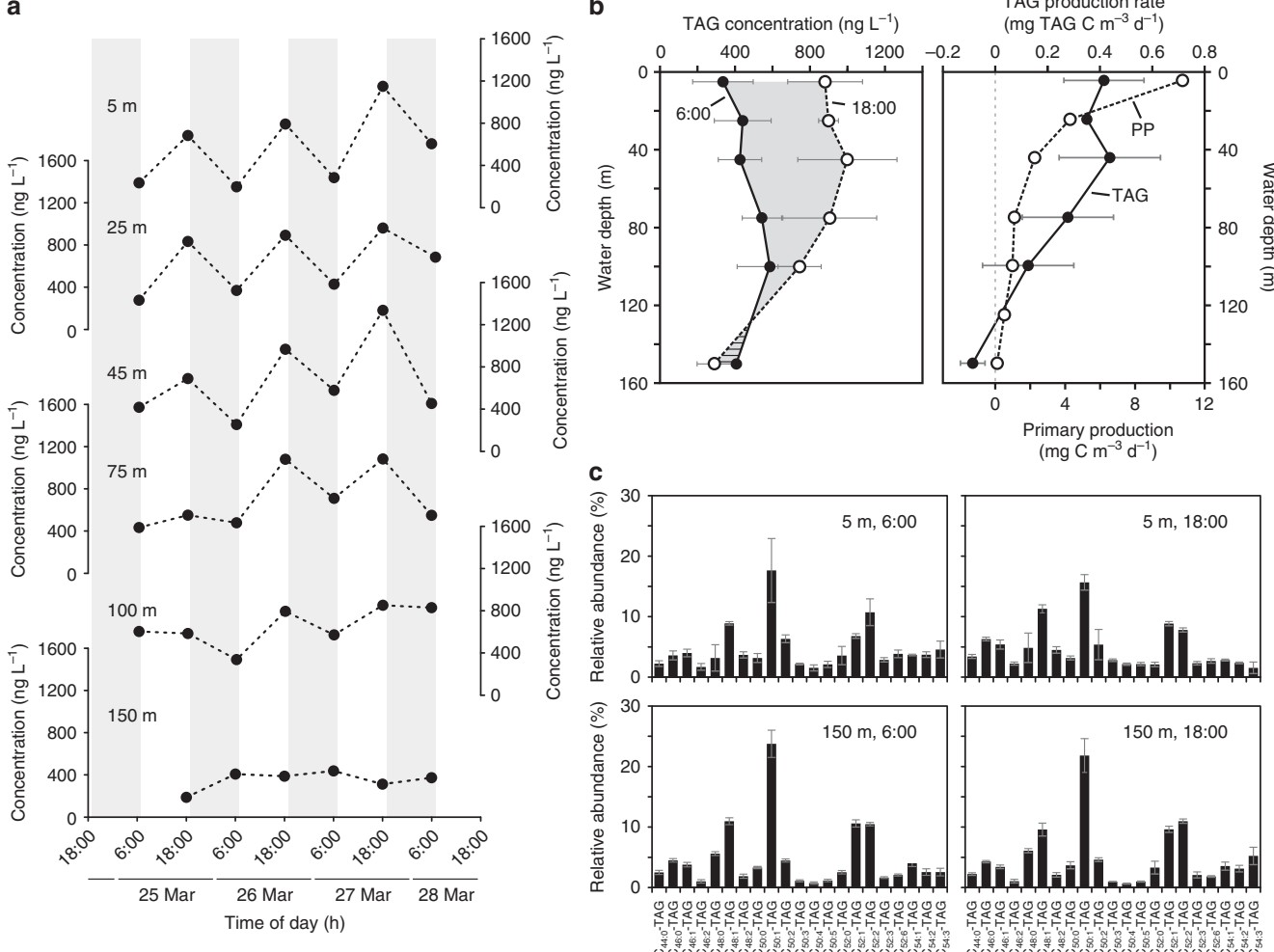

**Fig. 2** Diel patterns in triacylglycerols at different water depths. **a** Concentrations of triacylglycerols (TAGs) for dusk (~06:00 h, local time) and dawn (~18:00 h) over 4 day/night cycles (gray areas in indicate night) in March 2016 at different water depths at Station ALOHA. **b** Time-of-day average of depth profiles of TAGs and derived production rates (mg TAG C $m^{-3}$ $d^{-1}$) as well as primary production (PP, mg C $m^{-3}$ $d^{-1}$). **c** TAG composition averaged for time of day at 5 and 150 m water depth. Bars in **b** and **c** are the mean ± SD ($n \geq 3$)

**Table 2 Triacylglycerol (TAG) production rates and their contribution to overall daytime net primary production (PP)**

| | Day (m/d/y) | Water depth (m) | $^{14}$C-PP[a] (mg m$^{-3}$ d$^{-1}$) | 2–20 μm PP[a] (mg C m$^{-3}$ d$^{-1}$) | $^{14}$C-AR$_{night}$[b] (mg m$^{-3}$ d$^{-1}$) | TAG production[c] (mg C m$^{-3}$ d$^{-1}$) | %TAG C of $^{14}$C-PP | %TAG C of 2–20 μm PP |
|---|---|---|---|---|---|---|---|---|
| Summer 2015 | 7/27/2015 | 15 | 17.6 ± 0.4 | 5.4 ± 2 | 7.6 | 1.08 | 6.1 | 20 |
| | 7/28/2015 | 15 | 14.9 ± 2.4 | 2.9 ± 0.9 | 5.2 | 1.16 | 7.8 | 40 |
| | 7/29/2015 | 15 | 16.9 ± 1.5 | 4.3 ± 0.8 | 6.2 | 0.69 | 4.1 | 16 |
| | 8/1/2015 | 15 | 14.7 ± 1.2 | 3.2 ± 0.7 | 6.1 | 1.13 | 7.7 | 36 |
| Spring 2016 | 3/25/2016 | 25 | 5.8 ± 0.5 | n.d. | 1.8 | 0.42 | 7.3 | n.d. |
| | 3/26/2016 | 25 | 5.0 ± 0.7 | 2.7 ± 1.0 | 0.7 | 0.40 | 8.0 | 15 |
| | 3/27/2016 | 25 | 5.8 ± 0.5 | 3.3 ± 1.0 | 2.6 | 0.41 | 7.1 | 12 |

n.d. not determined
[a]Values represent the average of environmental triplicates ± SD ($n = 3$)
[b]Autotrophic respiration (AR$_{night}$) determined after Marra and Barber[76] from dark $^{14}$C incubations
[c]Daily net TAG production rates were calculated from the difference between the minimum and maximum concentrations observed during each 24-h day

the total NPSG community at 15 m in summer 2015 ranged between 14.7 ± 2.4 and 17.6 ± 0.4 mg C m$^{-3}$ d$^{-1}$, whereas primary production by nanophytoplankton (2–20 μm particle size fraction) was between 2.9 ± 0.9 and 5.4 ± 2 mg C m$^{-3}$ d$^{-1}$, which is as expected in the NPSG where *Prochlorococcus* is numerically dominant. The TAG production rate in surface waters (15 m) was 1.0 ± 0.2 mg C m$^{-3}$ d$^{-1}$ (mean ± SD; $n = 4$), which represented 6.4 ± 1.7% of total daytime net primary production and 28 ± 10% of primary production by nanophytoplankton. Similar results were found in the spring of 2016, when TAG synthesis averaged 0.41 ± 0.01 mg C m$^{-3}$ d$^{-1}$ ($n = 3$) and was 7.5 ± 0.5% of total daytime net primary production and 14 ± 2% of nanoeukaryotic primary production. An estimate of the global significance of TAGs by assuming that the TAG production rates from the spring are representative of mean rates throughout the year at Station ALOHA, integrating them through the photic zone, and extending them to the total area of subtropical gyres (see Supplementary Methods for calculation), indicates that the daily accumulation and depletion of TAGs could drive a carbon flux approaching 2.4 Pg C yr$^{-1}$. This is a low estimate of global TAG synthesis because this phenomenon likely also occurs outside the subtropical gyres in high latitude, equatorial, or coastal regions where primary production rates are higher and eukaryotic phytoplankton are more abundant. Thus, the total global percentage contribution of TAGs to primary production could be substantial, although light may limit carbon fixation in some of these other environments[47,48].

## Discussion

The diel cycle in TAG concentrations we observed in the NPSG is evidence that the biochemical composition and caloric content of eukaryotic phytoplankton changes profoundly throughout the course of a day. The availability of light energy cycles between extremes of darkness and abundance on a diel cycle, and phytoplankton buffer these variations by transforming this resource into stored chemical energy. Lipid molecules such as TAGs contain more chemical energy per unit mass than other classes of biochemicals, such as starch or carbohydrates[11], and the caloric content of phytoplankton is strongly correlated to their lipid content[49]. TAG molecules contain only atoms of hydrogen, carbon, and oxygen, elements that are readily available from water and dissolved carbon dioxide; consequently, TAG synthesis by phytoplankton does not incur a stoichiometric requirement for other elements, such as nitrogen, that are scarce in their environments. Thus, TAGs are an efficient molecular vehicle for storing chemical energy that is suited to the environmental conditions experienced by phytoplankton in the NPSG.

Due to the significant daily changes in the TAG content of eukaryotic phytoplankton, the timing of their mortality could have a large impact on ocean carbon cycling in the NPSG. Using the same approach as Platt and Irwin[49], we calculate that phytoplankton in the NPSG could contain as much as 40% more calories at dusk than at dawn (see Supplementary Methods for calculations). Thus, we hypothesize that eukaryotic phytoplankton would be more favorable prey for zooplankton at dusk than at dawn. Vertical mesozooplankton distributions in the NPSG and throughout the oceans show strong diel changes, with mesozooplankton ascending from the mesopelagic zone to the surface at dusk and descending at dawn. This daily migration, the largest coordinated movement of animals on the planet[50], balances two competing objectives: rapid growth through efficient feeding in phytoplankton-rich surface waters, and minimization of risk of mortality from visual predators. Traditionally, the cue for diel vertical migration of mesozooplankton was thought to be a relative change in visible light[51]. However, diel vertical migration was also observed in deep sea plankton and it was hypothesized that a precise biochemical clock could maintain the solar diurnal rhythms in deep sea plankton motions[52,53]. We hypothesize, based on the evidence in this study, that the caloric enrichment of phytoplankton at night could provide a strong evolutionary reinforcement for diel vertical migration by mesozooplankton. Life cycles of some mesozooplankton species depend on the nutritional quality of phytoplankton, as defined by lipid content[54].

In addition to potentially impacting the timing of zooplankton grazing, the diel fluctuations in TAG content of eukaryotic phytoplankton are likely tightly coupled to the fate of organic matter within the NPSG. If TAGs are respired entirely by phytoplankton, then their synthesis would not contribute to net primary production even though TAG production is a substantial fraction of gross production. Using a simple model, we estimate that around 70% of the TAGs produced during the day are consumed by eukaryotic phytoplankton themselves, whereas 30% is consumed by mortality processes (see Supplementary Methods, Supplementary Figure 11 and Supplementary Table 2 for model details). Grazing mortality diverts the energy in TAGs to higher trophic levels, whereas mortality via autolysis or viral lysis of TAG-rich eukaryotic phytoplankton results in the release of relatively hydrophobic, calorie-rich, and nutrient-poor molecules to the pool of dissolved organic matter that fuels heterotrophic bacteria. The consequences of these fluctuations for carbon export, dissolved organic matter composition, nutrient cycling, and ecosystem function are unconstrained. TAGs accumulate in distinct intracellular lipid droplets, the fate of which upon release from lytic cells is also unknown. Understanding the timing and the amount of organic matter transferred from TAG-producing phytoplankton to higher trophic levels and/or the dissolved organic carbon pool will elucidate the consequences of TAG cycling for the marine food web and carbon cycling.

## Methods

**Sample collection and lipid analysis by HPLC–MS.** Seawater samples were collected during R/V Kilo Moana cruises KM1513 (July/August 2015) and KM1605 (March 2016) near Station ALOHA (22°45'N, 158°00'W) in the oligotrophic North Pacific Subtropical Gyre using standard Niskin-type bottles. Triplicate samples were obtained from the bottles during the July cruise and single samples during the March cruise. Samples (~2 L) were filtered using vacuum filtration (ca. −200 mm Hg) onto 47 mm diameter 0.2 μm hydrophilic Durapore filters (Millipore) and immediately flash-frozen and stored at −196 °C until processing. Filters were extracted using a modified Bligh and Dyer extraction[55] with DNP-PE-$C_{16:0}$/$C_{16:0}$-DAG (2,4-dinitrophenyl phosphatidylethanolamine diacylglycerol; Avanti Polar Lipids, Inc., Alabaster, AL) used as an internal standard. Lipids were analyzed by reverse phase high performance liquid chromatography (HPLC) mass spectrometry (MS) on an Agilent 1200 HPLC (Agilent, Santa Clara, CA, USA) coupled to a Thermo Q Exactive Orbitrap high resolution mass spectrometer (ThermoFisher Scientific, Waltham, MA, USA). HPLC and MS conditions are described in Collins et al.[56] (modified after Hummel et al.[57]). In brief: 20 μL injections of sample were made onto a C8 Xbridge HPLC column (particle size 5 μm, length 150 mm, width 2.1 mm; Waters Corp., Milford, MA, USA). Eluent A was water with 1% 1 M ammonium acetate and 0.1% acetic acid. Eluent B was 70% acetonitrile, 30% isopropanol with 1% 1 M ammonium acetate and 0.1% acetic acid. Gradient elution was performed at a constant flow rate of 0.4 mL min$^{-1}$ using the following gradient: 45% A for 1 min to 35% A at 4 min, then from 25% A to 11% A at 12 min, then to 1% A at 15 min with an isocratic hold until 25 min, and finally back to 45% A for 5 min column equilibration. ESI source settings were: Spray voltage, 4.5 kV (+), 3.0 kV (−); capillary temperature, 150 °C; sheath gas and auxiliary gas, both 21 (arbitrary units); heated ESI probe temperature, 350 °C. Mass data were collected in full scan while alternating between positive and negative ion modes. For each MS full scan, up to three MS$^2$ experiments targeted the most abundant ions with $N_2$ as collision gas. A scan range of 100–1500 m/z was used for all modes in sequence. The mass spectrometer was set to a resolving power of 140,000 (FWHM at m/z 200) leading to an observed resolution of 75,100 at m/z 875.5505 of our internal standard, DNP-PE. Exact mass calibration was performed by weekly infusing a tune mixture. Additionally, every spectrum was corrected using a lock mass, providing real-time calibrations.

For the discovery, annotation, and putative identification of lipids in NPSG plankton, we used LOBSTAHS, an open-source lipidomics software pipeline that enables rapid identification of more than 20,000 individual lipids from an onboard library[56] based on adduct ion abundances and several other orthogonal criteria (see Supplementary Data 1 for identified and annotated mass features by LOBSTAHS). Lipids identified using the LOBSTAHS software were quantified from MS data after pre-processing with xcms[58] and CAMERA[59]. XCMS peak detection was validated by manual identification using retention time as well as accurate molecular mass and isotope pattern matching of proposed sum formulas in full-scan mode and tandem MS (MS$^2$) fragment spectra of representative compounds (Supplementary Figure 6).

**Lipid quantification.** Absolute lipid concentrations were calculated from peak areas of molecular ions in mass chromatograms using response factors from external standard curves. Individual response factors were determined by triplicate injection of a series of standard solutions in amounts ranging from 0.15 pmol to 40 pmol on column. Standards for phosphatidylglycerol (PG), phosphatidylethanolamine (PE) and phosphocholine (PC), diacylglyceryl trimethylhomoserine (DGTS), and DNP-PE all contained two palmitic acid acyl groups and were purchased from Avanti Polar Lipids, Inc., (Alabaster, AL; USA). The abundances of diacylglyceryl hydroxymethyl-trimethyl-β-alanine (DGTA) and diacylglycerylcarboxy-N-hydroxymethyl-choline (DGCC) were corrected using the DGTS standard. Purified sulfoquinovosyl diacylglycerol (SQDG) from spinach was purchased from Lipid Products (South Nutfield, UK). Response factors for quinones were determined using a ubiquinone ($UQ_{10:10}$) standard from Sigma-Aldrich (St. Louis, MO, USA). Triacylglycerols (TAGs) were quantified using a series of individual standards from Nu-Check-Prep, Inc. (Elysian, MN, USA). TAG response factors were based on the equivalent carbon number of each TAG after Holčapek et al.[60] (Supplementary Figure 7). To validate peak integration by the XCMS software, we compared concentrations of selected lipids obtained from auto integration with manual integration (Supplementary Figure 8). Statistical significance of diel (24-h) rhythmicity of lipid data was determined using the RAIN non-parametric test in R[20] with the default settings (independent).

Ion suppression was tested by further HPLC analysis of 30 selected samples spiked with the following deuterated standards: 1-pentadecanoyl-2-oleoyl(d7)-sn-glycero-3-phosphoethanolamine, 1,2-dipalmitoyl-sn-glycero-3-O-4'-[N,N,N-trimethyl(d9)]-homoserine, D-glucosyl-ß-1,1'-N-stearoyl-D-erythro-sphingosine-d5, 1-hexadecanoyl-2-(9Z-octadecenoyl)-sn-glycero-3-phospho-(1'-rac-glycerol-1',1',2',3',3'-d5), and 1,3(d5)-dihexadecanoyl-2-octadecanoyl-glycerol, purchased from Avanti Polar Lipids, Inc., (Alabaster, AL; USA). No systematic matrix ion suppression was observed in these selected samples (Supplementary Figures 9 and 10) and thus the remaining ≈120 samples were not similarly analyzed.

**Transcriptomic analysis.** Samples for transcriptomic analysis were collected during R/V Kilo Moana cruise KM1513 in tandem with the lipid samples.

Duplicate samples were sequentially filtered through a 100 μm nylon mesh pre-filter followed by a 142 mm 0.2 μm polycarbonate filter using a peristaltic pump. Filters were flash frozen in liquid nitrogen and subsequently stored at −80 °C until further processing. Filters were extracted using the ToTALLY RNA Kit (Invitrogen) with some modifications. Briefly, frozen filters were added to 50 mL falcon tubes containing 5 mL of denaturation solution and extraction beads (125 μL 100 μm zirconia beads, 125 μL 500 μm zirconia beads, and 250 μL 425–600 μm silica glass beads). In addition, to generate quantitative transcript inventories, a set of 14 internal mRNA standards were also added to the extraction buffer; these standards were synthesized according to Satinski et al.[61] with the exception that eight standards were synthesized with poly(A) tails to mimic eukaryotic mRNAs. Total extracted RNA was treated with DNase I (Ambion, New York, USA) and purified with DNase inactivation reagent (Ambion). Purified total RNA was used to construct TruSeq cDNA libraries according to the Illumina TruSeq® RNA Sample Preparation v2 Guide. Briefly, RNAs were poly(A)-selected, converted to cDNAs, and sheared to ~225 bp fragments which were used to prepare cDNA libraries for paired-end (2 × 150) sequencing using the Illumina NextSeq 500 sequencing platform (Illumina Inc., San Diego, CA). Raw Illumina data were quality controlled with trimmomatic v0.36[62] using the parameters MAXINFO:135:0.5, LEADING:3, TRAILING:3, MINLEN:60, and AVGQUAL:20, and matching read pairs were merged using flash v1.2.11[63] with parameters -r 150 -f 250 -s 25. FASTQ files were converted to FASTA and translated with seqret and transeq vEMBOSS:6.6.0.0[64] using Standard Genetic Code, keeping only potential ORFs with length> 40 amino acids.

Full-length DGAT1 and DGAT2 protein sequences from *Arabidopsis thaliana* (NP_179535; DGAT1) (NP_566952; DGAT2), *Saccharomyces cerevisiae* (NP_014888; DGAT2), *Phaeodactylum tricornutum* (43469; DGAT2), and *Ostreococcus tauri* (34368; DGAT2) were used as queries to identify orthologs in marine eukaryotic genomes and transcriptome assemblies available through Joint Genome Institute (JGI) and the Marine Microbial Eukaryote Transcriptome Sequence Project (MMETSP)[65], respectively, using BLASTp. These included both experimentally verified and putative DGAT1/2 orthologs previously described[38]. Sequences were clustered using usearch at an identity of 80% (DGAT1) or 50% (DGAT2) and aligned using MAFFT (Version 7) with the E-ISN-I algorithm[66]. The alignment was trimmed using trimAl v1.2[67] using the −gt .05 −resoverlap 0.5 −seqoverlap 50 options, and any sequences that did not contain the highly conserved DGAT1 histidine residue were removed[68]. The trimmed alignment file was converted to Phylip format, and the best-fit amino acid substitution matrix (LG), among-site rate heterogeneity model (gamma distribution with invariable sites), and observed amino acid frequency were determined using ProtTest 3 software[69,70]. Outgroups included several related sterol O-acyltransferase protein sequences. A bootstrapped ($n = 100$), best-scoring ($n = 10$) maximum-likelihood tree was built using RAxML Version 8[71,72].

To recruit environmental metatranscriptomic reads to the DGAT1/2 reference alignments, an hmm profile was constructed from each reference alignment using hmmbuild followed by transcript identification and alignment using hmmsearch (parameters: -T 35 −incT 35) and hmmalign, respectively, using the HMMER package v3.1b2[73]. Taxonomy was assigned to each environmental sequence using pplacer v1.1.alpha19-0-g807f6f3[74] based on the read placement with the best maximum likelihood score to the reference tree (parameters: --keep-at-most 1). Read counts for each edge were normalized by recovery of internal mRNA standards to estimate natural transcript abundance per liter of seawater[61]. To assess transcription of additional protein families, we obtained hmm profiles from Pfam 31.0[75] for the protein families Acyltransferase (PF01553), Triglyceride lipase (PF01764), Lipase maturation factor (PF06762), and Acyl-CoA-binding protein (PF00887). These hmm profiles were then used with hmmsearch using the same parameters listed above. Statistical significance for diel periodicity of transcripts was tested using the RAIN package in R[20].

**Primary production (PP) measurements.** Standard Hawaii Ocean Time-series protocols were used to measure in vitro dawn to dusk PP via the $^{14}C$ radiotracer method[46]. Water samples were collected from 25 m before dawn and incubated on deck from local dawn to dusk in incubators screened to 55% of surface irradiance and plumbed with flow through surface seawater for temperature regulation. Rate measurements in light bottles were corrected to remove the rate measured in parallel dark bottles, which were on average $4 \pm 1\%$ ($n = 14$) of light values. To determine night-time respiration, samples were incubated for 12 h in the light with $^{14}C$-labeled inorganic carbon and then incubated for a further 12 h in the dark as described in Marra and Barber[76].

The potential contributions of the 2–20 μm size class to overall PP was calculated using an optical approach to measure particle size distributions (PSD) as described in White et al.[77]. Briefly, an in situ laser diffractometer (LISST-×100, Laser In Situ Scatterometer/Transmissometer, Sequoia Scientific Inc., hereafter simply LISST), was used to make continuous underway measurements of the PSD. Water was supplied from the ship's uncontaminated seawater, which drew seawater from ~7 m below the waterline of the forward hull. For the first 10 min of every hour incoming seawater was passed through a 0.2 μm filter. This dissolved blank was used to correct whole water signals collected in the succeeding 50 min of each hour. Particle volume estimated via a spherical inversion of forward scatter in 32 log-spaced bins[77,78] was converted to carbon using the allometric scaling of

Menden-Deuer and Lessard[79] for non-diatom protistan plankton. Finally, the production of the 2–20 µm size class, which showed distinct diel cycles with a minimum at dawn and a maximum at dusk (Supplementary Figure 1), was calculated as the amplitude of this daily increase divided by the length of the photoperiod just as was done for TAG production.

The PP vertical profile for March 2016 was calculated by discretization of the below equation at the standard sampling depths of Station ALOHA: P (mg C m$^{-3}$) = 12,000 × PAR × [chl $a$] × $a^*$ × $\Phi c$, where 12,000 converts mol of carbon (C) to mg C, mean chlorophyll-specific absorption coefficient based on HPLC ($a^*$, 0.026) is as reported by Letelier et al.[80] for summer months and $\Phi c$, the quantum yield of photosynthesis [mol C mol quanta$^{-1}$] was parameterized from $^{14}$C-based photosynthesis-irradiance curves as per Li et al.[81].

## Data availability

Sequence data are available in the NCBI BioProject database (accession number PRJNA396249). All other relevant data are available from the corresponding author upon request.

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

## Acknowledgements

We are grateful to the officers and crew of the *R/V Kilo Moana* cruises KM1513 and KM1605. We further thank Allison Coe in the Chisholm Lab at MIT for providing *Prochlorococcus* biomass. Thanks to four anonymous reviewers whose comments greatly improved this manuscript. This work was supported by a grant from the Simons Foundation, and is a contribution of the Simons Collaboration on Ocean Processes and Ecology (SCOPE award # 329108, B.A.S.V.M.). K.W.B. was further supported by the Postdoctoral Scholarship Program at Woods Hole Oceanographic Institution & U.S. Geological Survey.

## Author contributions

K.W.B., lipidomics, data analysis, and paper writing; B.A.S.V.M, study design, paper writing; J.R.C., lipidomics and data analysis; B.P.D., R.D.G. and E.V.A., transcriptomics and data analysis. A.E.W., primary productivity measurements. H.F.F., lipidomics and data analysis; D.J.R. and J.E.O., onboard sampling; P.C., *Pelagibacter* biomass provision; All authors provided editorial comments on the manuscript.

## Additional information

**Competing interests:** The authors declare no competing interests.

