## [Peer Review File · Nature Communications]

Reviewers' comments:

Reviewer #1 (Remarks to the Author):

The authors submit a revised version of the paper 'Daily changes in phytoplankton lipidomes reveal mechanisms of energy storage in the open ocean'.

They have responded to my original comments well and now present a more robust and considered analysis of their data. Particularly as to the novelty and significance of the findings. They include more caveats relating to their presented proposals and a more rigorous analysis of previous data. While this makes the paper scientifically sound but potentially reduces its novelty/significance.

The authors now also present a model of the potential fate of carbon stored as lipid which gives a better context to the paper and improves the papers value to the community however much of the significance (i.e. in migration/carbon cycle) is conjecture.

I have minor corrections below:

Line 50 states that it is know that prokaryotes store glycogen and eukaryotes store TAGS, and that (ref 21 form the eastern tropical north pacific) that eukaryotic phytoplankton are the main source of TAG in the ocean. However, on Line 103 the authors present 'our hypothesis that eukaryotic phytoplankton dominate the TAG signal in the NPSG' – I don't think this should be presented as the authors hypothesis – only that they are confirming current thinking.

Line 65 – 'persist in the face of numerical dominance by Prochlorococcus' – this comment is a little over the top - they simply have different life strategy on many levels (cell size, pigments, nutrient transporters, storage vacuoles, mixotrophy) – accumulation of TAG is one.

Line 65 'play an outsize role in the carbon cycle' – also a little over the top. Its depends on carbon fixation, rate of export, grazing rate etc etc. Possibly use 'a previously uncharacterised role in the carbon cycle'

Line 205 – 'We posit instead that the diel signal in biosynthesis of TAGs is driven directly by sunlight availability'. I agree that their data supports a light driven signal but this is what you would expect as nutrient concentrations in the ocean tend to vary over different temporal/special scale than a diel? So yes, the NASG is nutrient limited (potentially including high TAG accumulation per cell than and high nutrient region), but this signal would be seasonal not diel. Or maybe change the line to ...

'We posit instead that the magnitude of the diel signal in biosynthesis of TAGs with depth is driven directly by sunlight availability'

Reviewer #2 (Remarks to the Author):

The authors argue that daily cycles of triacylglycerides, but not membrane lipids, of specific species give relevant new insights into overall carbon flux in the ocean. The reviewer fails to see the significance of this observation (even if analytical measurements were correct). The authors argue that cell numbers (measured by the relative stability of membrane lipid levels) remain constant, while TAG levels varied. To the reviewer, such observation would only reflect biosynthesis of storage lipids during the day, and use of TAG energy in beta-oxidation during the night, to fuel overall energy needs of the cell. To the reviewer, this is not a very noteworthy finding that would be useful for greater implications in overall ocean energy homeostasis or carbon dioxide use.

With respect to analytical chemistry, the authors have not revised their methods or improved analyses in relation to the first manuscript submission. The authors argue that similar papers using this method had been published elsewhere. Such argument is not a scientifically valid rebuttal. Absolute quantifications in LC-MS must be based on internal standards, preferably, by stable isotope labeled versions of the quantified compounds. A single non-lipid standard with external calibration response curves is inadequate, because calibration response curves may differ greatly under matrix effects. The reviewer advises the authors to refer to methods published in the authoritative Journal of Lipid Research. Further internal standards are available by AvantiPolar Lipids, who even make new standards based on community input. If no labeled internal standards are available, surrogate standards can be used, or (if no other way is found), standard additions to the matrix can be used. For these arguments, the absolute quantifications given by the authors cannot be evaluated here. The reviewer is also confused by the use of 'fold-change', with 1.3-fold changes in TAGs and 0.27 fold changes in membrane lipids. The reviewer would expect here molar changes (per million cells, or per liter of seawater) - the fold changes invite misinterpretations as given here.

The authors are also very vague about lipid identifications. They claim "Lipids putatively identified using the LOBSTAHS"... The reviewer does not understand the wording 'putatively identified'. Lipids are commonly identified by MS/MS analyses. The authors here appear to rely on retention time - m/z features, which heavily relies on adduct formation. It is also unclear how many lipid isomers or isobars were co-eluting or in which way isotopes and adducts of some lipids would impact the quantification of other co-eluting lipids. Here, the use of LC with ion mobility and subsequent mass spectrometry would give clarity.

Reviewer #3 (Remarks to the Author):

This paper presents a compelling and unique dataset that documents diel oscillations in triacylglycerol abundances among small marine eukaryotic phytoplankton. I did have some concerns about the interpretation of the diel TAG abundance cycling in terms of primary production, and in the use of the metatranscriptome data to make inferences about TAG metabolism.

I. 26: "unconstrained" is an inappropriate word choice here; there is, in fact, a wealth of information on this topic (mostly from cultured isolates), though of course there is still much more to investigate — in particular, as the authors point out, in the field.

I. 30: "Diel oscillations in TAG concentration comprise $23 \pm 11\%$ of primary production by phytoplankton" — this statement is at best misleading: it's more like 23% of primary productivity in the 2-20 μm size fraction (cf. Table 1), but closer 7% of actual "primary production by phytoplankton".

The manuscript repeatedly refers to TAG synthesis as contributing to "net primary production" yet also attributes around 70% of the nighttime decline in TAG abundance to respiratory consumption by the phytoplankton that made them during the daytime. This is the very definition of the distinction between net and gross primary production: net primary production is equal to gross primary production minus respiration by primary producers. So the calculations presented for daytime net TAG synthesis (and I would encourage the authors to qualify statements about "net TAG production rates", e.g. line 224, as specifically daytime rates) really apply to gross, not net, primary productivity (at least, on timescales longer than half a day). The contribution of TAGs to net primary productivity might be better described as the long-term average TAG concentration (say $\sim 2\mu\text{g/liter}$) divided by the standing stock of primary producer biomass (which is not given,

but seems must have been calculated in the procedure described in l. 395-408).

The DGAT transcriptome data is not put in context of the community composition. A reasonable null hypothesis is that clade-specific TAG synthesis simply tracks cell abundance and/or relative contribution to primary production. Does the transcriptome data support or refute this hypothesis? Do some taxa appear to contribute disproportionately to TAG production as compared to what would be expected from, for example, reads of housekeeping, photosynthesis and/or carbon fixation genes? Adding this sort of comparative analysis would make the transcriptome data more than just a "whodunit" inventory, and aid in the interpretation of other datasets.

I also find the lack of temporal correspondence between DGAT transcripts and TAG abundances to be an important result, and worth highlighting (personally, I would incorporate Supplementary Fig 5 into the main Fig 1). Too often metatranscriptome data are naively interpreted as a real-time readout of metabolism, despite abundant evidence to the contrary. This dataset is a nice counterexample. I am a bit concerned, though, about the leap to saying that "TAG abundance in the NPSG might be regulated through intracellular consumption instead of production" (l. 160) in the absence of data concerning translational, post-translational or flux-based regulation of TAG synthesis. Such data may not exist for relevant taxa, but since the authors have access to full metatranscriptomes from these samples, the case for what is going on with TAG production could be made much stronger by at least looking at it in a pathway context, by also looking at abundance patterns of glycerol phosphate pathway and phospholipid synthesis genes. Though the diel oscillation in TAG abundance doesn't appear to be transcriptionally regulated at the DAG-to-TAG step, regulation of an upstream or branching step in the pathway might yield useful insights. If none of the genes in the larger pathway show diel periodicity in transcript abundance, the case for post-transcriptional and/or degradational control of TAG abundance would be much strengthened.

Fig 2: The phylogenetic tree is really not especially relevant to the argument of the paper, and is mostly illegible anyway. The important data — the relative transcript counts apportioned among various groups — might be better represented by a table. The tree could go in a supplemental figure.

Reviewer #4 (Remarks to the Author):

The manuscript "Daily changes in phytoplankton lipidomes reveal mechanisms of energy storage in the open ocean" (NCOMMS-18-12389-T) is an improved version of the original submission as several areas of concern have been addressed. The manuscript reads well and generally presents an interesting idea – that the accumulation and consumption of lipids (TAGs) in natural marine phytoplankton communities is a poorly characterized, but important component of the marine carbon cycle.

While the manuscript is improved, most of the comments on the previous version were not well-addressed. Specifically:

- (1) Past work has been cited, but there is little attempt to quantitatively compare the results presented here with past, primarily laboratory, work.
- (2) The gyre calculation is frustratingly basic and does not consider the measured vertical variability of TAG production, the temporal/seasonal variability of primary production (and likely of TAGs), that TAG production is likely highly dependent on nutrient concentrations.
- (3) Perhaps minor, but frustrating for this reviewer...not including the primary production temporal and vertical variability in Fig 3 (as well as the POC, which may/may not have been measured on a diel basis) limits the interpretation for the reader – especially when this is a stated outcome (lines 77-79) of this paper. Even a supplementary figure would be welcome.

Reviewer #5 (Remarks to the Author):

Title: Daily changes in phytoplankton lipidomes reveal mechanisms of energy storage in the open ocean

This review refers to the lipidomic analyses included in the work

In this manuscript, Becker and co-workers used a lipidomics to evaluate the effect of sunlight in phytoplankton, particularly in TAG.

The lipidomics approach used in this manuscript includes the initial step of lipid extraction, using a modified Blight and Dyer protocol and an internal standard was added prior to the extraction to evaluate the lipid recovery. Reverse phase LC-MS was used to identify and quantify the lipidome of phytoplankton.

The absolute quantitation of each lipid molecular species was calculated from peak areas of the molecular ions in mass chromatograms using response factors from external standard curves and response factors. However no internal standards of each lipid class were added.

Recommendation for the revision to the manuscript:

Authors reported that they did absolute quantitation; however absolute quantitation requires the use of internal standards for each lipid class. There are several lipids standards in Avanti, either with odd chain fatty acids or deuterated lipids for all the phospholipid classes. In the case of betaines or glycolipids, you can choose one PL class with similar ionization efficiency. For example, you can use the PC standard, as internal standard for betaine class.

This is quite important particularly for reverse phase LC-MS lipid analysis, since in these cases there are co elution of ions from different classes, since elution of lipid species is quite dependent on the fatty acyl chains rather than the polar head group polarity. This contribute to ion suppression in the case of co elution of classes with different ionization capabilities, due to different ionization efficiencies for the lipid classes. This cannot be predicted with external calibration curves, that did not take into account the suppression effects in a lipid mixture. If internal standard are no used you cannot talk about absolute quantitation but rather relative quantitation.

Other suggestions

- 1) Information about the type of HPLC column used must be added.
- 2) Identification of the lipid species were only based on open source software? Did you confirm the identity of the lipid species by the analysis of raw data: LC-MS and MS/MS?
- 3) Include in the supplementary file a list of all the lipid species identified and quantified, with observed mass, calculated mass and error (ppm). Provide information of fatty acyl composition, if possible.
- 4) Include in supplementary file one MS and one MS/MS spectra representative of each class identified.
- 5) Why did you not evaluate the variation of the lipid profile for the other lipid classes other than TAG?

Reviewers' comments:

Reviewer #1 (Remarks to the Author):

The authors submit a revised version of the paper 'Daily changes in phytoplankton lipidomes reveal mechanisms of energy storage in the open ocean'.

They have responded to my original comments well and now present a more robust and considered analysis of their data. Particularly as to the novelty and significance of the findings. They include more caveats relating to their presented proposals and a more rigorous analysis of previous data. While this makes the paper scientifically sound but potentially reduces its novelty/significance.

We appreciate Review #1's comments on our revised manuscript. We have continued to make improvements to our paper to increase its robustness. In addressing the comments of Reviewer #1 and the other Reviewers, we believe we have established new, novel insights on lipid metabolism in phytoplankton communities.

The authors now also present a model of the potential fate of carbon stored as lipid which gives a better context to the paper and improves the papers value to the community however much of the significance (i.e. in migration/carbon cycle) is conjecture.

I have minor corrections below:

Line 50 states that it is know that prokaryotes store glycogen and eukaryotes store TAGS, and that (ref 21 form the eastern tropical north pacific) that eukaryotic phytoplankton are the main source of TAG in the ocean. However, on Line 103 the authors present 'our hypothesis that eukaryotic phytoplankton dominate the TAG signal in the NPSG' – I don't think this should be presented as the authors hypothesis – only that they are confirming current thinking.

We have deleted "our hypothesis".

Line 65 – 'persist in the face of numerical dominance by Prochlorococcus' – this comment is a little over the top - they simply have different life strategy on many levels (cell size, pigments, nutrient transporters, storage vacuoles, mixotrophy) – accumulation of TAG is one.

We have reworded this sentence. We now state only that TAGs "contribute to the ability of eukaryotic phytoplankton to compete with cyanobacteria".

Line 65 'play an outsize role in the carbon cycle' – also a little over the top. Its depends on carbon fixation, rate of export, grazing rate etc etc. Possibly use 'a previously uncharacterised role in the carbon cycle'

Changed according to the reviewer's suggestion.

Line 205 – 'We posit instead that the diel signal in biosynthesis of TAGs is driven directly by sunlight availability'. I agree that their data supports a light driven signal but this is what you would expect as nutrient concentrations in the ocean tend to vary over different temporal/special scale than a diel? So yes, the NASG is nutrient limited (potentially including high TAG accumulation per cell than and high nutrient region), but this signal would be seasonal not diel. Or maybe change the line to ... 'We posit instead that the magnitude of the diel signal in biosynthesis of TAGs with depth is driven directly by sunlight availability'

Changed according to the reviewer's suggestion.

Reviewer #2 (Remarks to the Author):

The authors argue that daily cycles of triacylglycerides, but not membrane lipids, of specific species give relevant new insights into overall carbon flux in the ocean. The reviewer fails to see the significance of this observation (even if analytical measurements were correct). The authors argue that cell numbers (measured by the relative stability of membrane lipid levels) remain constant, while TAG levels varied. To the reviewer, such observation would only reflect biosynthesis of storage lipids during the day, and use of TAG energy in beta-oxidation during the night, to fuel overall energy needs of the cell. To the reviewer, this is not a very noteworthy finding that would be useful for greater implications in overall ocean energy homeostasis or carbon dioxide use.

We respectfully disagree with Reviewer #2 about the noteworthiness of our finding. The plankton communities in the ocean, which we sampled, are a complex mixture of many thousands of different photosynthetic and heterotrophic microbial species. The expected cellular rate of a biochemical pathway based on culture studies of an organisms may not be manifest when the same organism is living in its natural environment experiencing stress from both nutrient scarcity and agents of mortality. One need look no further for an example of this than the rate of photosynthesis! That a single pathway, TAG biosynthesis, would emerge from this complexity with such exquisite rhythmicity, and account for so much organic carbon, is totally unexpected. We know of no other study showing a similar result.

With respect to analytical chemistry, the authors have not revised their methods or improved analyses in relation to the first manuscript submission.

Reviewer #2's statement that we "have not revised their methods" is not accurate. In Reviewer #2's comments on the original version of our manuscript they expressed doubts about the accuracy of automated peak integration we use. In response, we re-analyzed a part of the TAG results by integrating each peak by hand (Supplementary Figure 2); this represented many days of effort, which in the end showed, as we posited in our original manuscript, that the automated lipidomics workflow we use is accurate for quantification.

The authors argue that similar papers using this method had been published elsewhere. Such argument is not a scientifically valid rebuttal.

We disagree with this statement as well. The purpose of publishing exhaustive descriptions of analytical methods in journals specializing in such work, as was the case in the paper in question that we published in *Analytical Chemistry* (Collins et al., 2016), is to establish methods so that others can use the methods in subsequent studies without re-establishing the validity of the method. Most of the reviewer's concerns about our method, were rigorously addressed in that paper. Clearly we cannot expect Reviewer #2 to know all of the details of Collins et al., but, on the other hand, we do not think that we should be expected to re-litigate all of the details of that method in our current paper to *Nature Communications*. This stated, we have made numerous changes in the manuscript in an attempt to meet Reviewer #2 halfway.

Absolute quantifications in LC-MS must be based on internal standards, preferably, by stable isotope labeled versions of the quantified compounds. A single non-lipid standard with external calibration response curves is inadequate, because calibration response curves may differ greatly under matrix effects. The reviewer advises the authors to refer to methods published in the authoritative *Journal of Lipid Research*. Further internal standards are available by AvantiPolar Lipids, who even make new standards based on community input. If no labeled internal standards are available, surrogate standards can be used, or (if no other way is found), standard additions to the matrix can be used. For these arguments, the absolute quantifications given by the authors cannot be evaluated here.

Matrix effects in electrospray ionization of lipids tend to be most problematic in very rich samples (e.g., cultures) or samples with complex, often molecularly-uncharacterizable, co-extracting molecules (e.g., sediments). In our case, we are working with very dilute samples composed of >50% living biomass, composed of >90% molecularly characterizable molecules.

Nonetheless, as Reviewer #2 suggested, we have now conducted tests of matrix effects using a series of deuterated standards. We now explicitly show that matrix effects did not affect our quantification, and we have included these results in the Supplementary Information of the revised manuscript.

We chose 30 samples from our archive (17 from the 2015 and 13 from the 2016 data set) and spiked them with five deuterated standards including a TAG and a betaine lipid (DGTS) and compared the peak areas in the spiked samples with those from pure standard analysis (Fig. 1 below). On the x-axis are peak areas of the deuterated internal standards in these samples, and on the y-axis are the peaks areas of the deuterated standards alone in an identical amount of solvent. All 5 standard fall on the 1:1 line and thus, matrix effects had only a very little effect on lipid ionization and detection in our samples. This becomes even more apparent when the lipid concentrations in individual samples are corrected for ion suppression and are compared to the original concentrations (Fig. 2 below). It was decided, based on the results shown in Figures 1 and 2, that reanalyzing all the samples was not necessary, particularly considering the preciousness of our research funding and competing demands for instrument time associated with having undergraduates in the lab this summer.

Fig 1. Signal response comparisons for deuterated standards added to lipid extracts and deuterated standards measured alone. Cross plots of peak areas of deuterated standards versus deuterated standards added to lipid extracts from samples collected in spring 2016 (a) and summer 2015 (b). Error bars represent the standard deviation of averaged values with $n = 13$ for a and $n = 17$ for b. PE, phosphatidylethanolamine; DGTS, diacylglyceryltrimethyl-homoserine; PG, phosphatidylglycerol; TAG, triacylglycerol. The vertical error bars are smaller than the symbols.

Fig. 2. Effects of ion suppression on TAG concentrations from deuterated standard addition experiments. a, TAG concentration of triplicate samples from July 2015 before (closed symbols) and after (open symbols) correction for ion suppression. **b,** TAG concentration in samples from March 2016 before (black circles) and after (yellow circles) correction for ion suppression.

The reviewer is also confused by the use of 'fold-change', with 1.3-fold changes in TAGs and 0.27 fold changes in membrane lipids. The reviewer would expect here molar changes (per million cells, or per liter of seawater) - the fold changes invite misinterpretations as given here.

All data has been normalized to one liter of seawater (see Figure 1a). Figure 1b showed the average relative change for one day using these values. There are many fold-change calculation formulas. The fold change (relative change) that we had used is defined as $A/B-1$, which has the advantage that no change results in the value 0, 100% increase results in 1, and 100% decrease results in -1. We understand the confusion with this expression and now show the "original" calculation for fold change (A/B) in the figure instead. The main text has been changed accordingly.

There are number of practical reasons for using mass per liter. First and foremost, the number of cells responsible for TAG synthesis is unknown. Second, the number of liters is known, which allows us to scale our measurements and estimate the total global synthesis rate of TAGs.

The authors are also very vague about lipid identifications. They claim "Lipids putatively identified using the LOBSTAHS"... The reviewer does not understand the wording 'putatitvely identified. Lipids are commonly identified by MS/MS analyses. The authors here appear to rely on retention time - m/z features, which heavily relies on adduct formation. It is also unclear how many lipid isomers or isobars were co-eluting or in which way isotopes and adducts of some lipids would impact the quantification of other co-eluting lipids. Here, the use of LC with ion mobility and subsequent mass spectrometry would give clarity.

We have deleted "putatively".

To validate identifications by the LOBSTAHS software package, we manually check retention time as well as accurate molecular mass and isotope pattern matching of proposed sum formulas in full-scan mode and tandem MS (MS^2) fragment spectra of a number of representative compounds. We have now included this information in the manuscript and show a supplementary figure of mass spectra of representative compounds for each lipid group that is discussed in the paper. This information was also requested by Reviewer #5.

The LOBSTAHS software explicitly addresses the problems of isobars and isomers through the adduct hierarchy at the heart of the method. In order for a lipid feature to be ambiguously assigned, it would have to have the same mass (within 5 ppm), the same retention time window, and the same quantitative adduct formation. Rare cases of ambiguity, are coded by LOBSTAHS and then manually assessed through MS_2 . In the case of TAGs, there are no other isomers known to be abundant in the ocean. Upon ionization, TAGs do form $[M+NH_4]^+$ adducts, but because of TAGs' unique chemical properties, they elute later in our chromatographic method than almost all other lipids.

We appreciate where Reviewer #2 is coming from though, and again acknowledge that most readers of *Nature Communications* will not be familiar with LOBSTAHS. In the revised manuscript we have extended the methods section and supplementary information to make the methods clearer. We have also included a wealth of additional information about the molecules we identified, including MS^2 spectra. Lipid isomers, such as lipids with different double bond positions in the fatty acyl chains, can eventually not be separated, but this will not affect our quantification because we do not distinguish isomers in this paper.

Reviewer #3 (Remarks to the Author):

This paper presents a compelling and unique dataset that documents diel oscillations in triacylglycerol abundances among small marine eukaryotic phytoplankton. I did have some concerns about the interpretation of the diel TAG abundance cycling in terms of primary production, and in the use of the metatranscriptome data to make inferences about TAG metabolism.

We very much appreciate Reviewer #3's continued support of our manuscript. We interpret the question raised by this reviewer as a good sign: the point of a paper in *Nature Communications* is to stimulate the community into thinking about new findings and posing the important follow-up questions. To the best of our ability, we provided answers to most of the questions below but were unable, for reasons explained, to incorporate all of them in the manuscript.

I. 26: "unconstrained" is an inappropriate word choice here; there is, in fact, a wealth of information on this topic (mostly from cultured isolates), though of course there is still much more to investigate — in particular, as the authors point out, in the field.

We have changed "unconstrained" to "poorly elucidated".

I. 30: "Diel oscillations in TAG concentration comprise $23 \pm 11\%$ of primary production by phytoplankton" — this statement is at best misleading: it's more like 23% of primary productivity in the 2-20 μm size fraction (cf. Table 1), but closer 7% of actual "primary production by phytoplankton".

We qualified this statement in the previous sentence where we narrowed the discussion to "eukaryotic nanophytoplankton." We see the potential confusion here, and we have changed "phytoplankton" to "eukaryotic nanophytoplankton."

The manuscript repeatedly refers to TAG synthesis as contributing to "net primary production" yet also attributes around 70% of the nighttime decline in TAG abundance to respiratory consumption by the phytoplankton that made them during the daytime. This is the very definition of the distinction between net and gross primary production: net primary production is equal to gross primary production minus respiration by primary producers. So the calculations presented for daytime net TAG synthesis (and I would encourage the authors to qualify statements about "net TAG production rates", e.g. line 224, as specifically daytime rates) really apply to gross, not net, primary productivity (at least, on timescales longer than half a day).

This is exactly correct. On a 24-hour basis, the respiration of TAGs at night contributes to the difference between gross and net primary productivity. The comparisons we are making in the paper are to 12-hour, daylight (i.e. sunrise until sunset) ^{14}C primary production measurements. Since the increase in the concentration (mol/L) of TAGs during the day is equal to the decrease in the concentration of TAGs at night, then TAGs do not contribute to gross production per liter. However, the TAG concentration does not come to zero every morning, and thus TAGs likely contribute to gross production at the scale of individual cells. This is a nuance that is beyond

the scope of a first paper on ocean TAG synthesis for a journal like *Nature Communications*. Our main reason for comparing to net primary production is to illustrate the global scale of the carbon flux through the TAG reservoir; recall, our measurements of the TAG synthesis are methodologically independent from ¹⁴C primary production measurements.

The contribution of TAGs to net primary productivity might be better described as the long-term average TAG concentration (say ~2µg/liter) divided by the standing stock of primary producer biomass (which is not given, but seems must have been calculated in l. 395-408).

This is an intriguing idea, which we gave some thought to while preparing our manuscript. The problem is that we do not know the fraction of POC that is composed of short-lived biomass versus long-lived detritus. However, the point is well taken. If we assume that total POC (40 micrograms C L⁻¹, during July/August cruise) is three quarters living biomass and compare the average TAG concentration (1.8 µg C L⁻¹), we arrive at 6%. This is very close to the average of 6.4% we report in the manuscript. Hopefully this calculation addresses this concern of Reviewer #3, we just don't feel comfortable including it in the manuscript.

The DGAT transcriptome data is not put in context of the community composition. A reasonable null hypothesis is that clade-specific TAG synthesis simply tracks cell abundance and/or relative contribution to primary production. Does the transcriptome data support or refute this hypothesis? Do some taxa appear to contribute disproportionately to TAG production as compared to what would be expected from, for example, reads of housekeeping, photosynthesis and/or carbon fixation genes? Adding this sort of comparative analysis would make the transcriptome data more than just a "whodunit" inventory, and aid in the interpretation of other datasets.

Again, this is another great idea. The type of data needed for this are available, but are at the heart of another manuscript by a different set of co-authors, and we are not able to incorporate them into our paper. However, for the edification of reviewer #3: dinoflagellates and haptophytes contribute approximately 50% and 20%, respectively, of total mRNA, and approximately 30% and 5%, respectively, of total rRNA (based on 18S). This is similar to the 46% and 25% of the DAGAT reads we report for dinoflagellates and haptophytes, respectively. So it would seem that the DAGAT scales with community structure. HOWEVER, extending this interpretation to the fraction of TAG synthesis is fraught with uncertainty.

I also find the lack of temporal correspondence between DGAT transcripts and TAG abundances to be an important result, and worth highlighting (personally, I would incorporate Supplementary Fig 5 into the main Fig 1). Too often metatranscriptome data are naively interpreted as a real-time readout of metabolism, despite abundant evidence to the contrary. This dataset is a nice counterexample. I am a bit concerned, though, about the leap to saying that "TAG abundance in the NPSG might be regulated through intracellular consumption instead of production" (l. 160) in the absence of data concerning

translational, post-translational or flux-based regulation of TAG synthesis. Such data may not exist for relevant taxa, but since the authors have access to full metatranscriptomes from these samples, the case for what is going on with TAG production could be made much stronger by at least looking at it in a pathway context, by also looking at abundance patterns of glycerol phosphate pathway and phospholipid synthesis genes. Though the diel oscillation in TAG abundance doesn't appear to be transcriptionally regulated at the DAG-to-TAG step, regulation of an upstream or branching step in the pathway might yield useful insights. If none of the genes in the larger pathway show diel periodicity in transcript abundance, the case for post-transcriptional and/or degradational control of TAG abundance would be much strengthened.

We have incorporated the DGAT diel variability into Fig. 1.

We have now looked for hits for pathways upstream and downstream of TAGs in PFAM, and found that only a relatively small number of pathways were represented in the major eukaryotic phytoplankton groups. With the exception of a relatively scarce acyltransferase and an acyl-CoA-binding protein in haptophytes, none showed diel periodicity. We have included this information in a supplementary table (Supplementary Table 1).

Fig 2: The phylogenetic tree is really not especially relevant to the argument of the paper, and is mostly illegible anyway. The important data — the relative transcript counts apportioned among various groups — might be better represented by a table. The tree could go in a supplemental figure.

We have deleted the figure and now show a table instead. A detailed phylogenetic tree is shown in Supplementary Fig. 10.

Reviewer #4 (Remarks to the Author):

The manuscript “Daily changes in phytoplankton lipidomes reveal mechanisms of energy storage in the open ocean” (NCOMMS-18-12389-T) is an improved version of the original submission as several areas of concern have been addressed. The manuscript reads well and generally presents an interesting idea — that the accumulation and consumption of lipids (TAGs) in natural marine phytoplankton communities is a poorly characterized, but important component of the marine carbon cycle.

We appreciate Reviewer #4's continued enthusiasm about our manuscript.

While the manuscript is improved, most of the comments on the previous version were not well-addressed. Specifically:

We certainly thought that we had addressed Reviewer #4's comments, and thank them for their patience as we make another attempt.

(1) Past work has been cited, but there is little attempt to quantitatively compare the results presented here with past, primarily laboratory, work.

We have now added a quantitative comparison with the magnitude of the TAG increase is in a culture experiment with both a haptophyte and a dinoflagellate (e.g., Lacour et al., 2012; Chen et al., 2011).

(2) The gyre calculation is frustratingly basic and does not consider the measured vertical variability of TAG production, the temporal/seasonal variability of primary production (and likely of TAGs), that TAG production is likely highly dependent on nutrient concentrations.

We have now use our vertical profile of TAG production from the 2016 spring cruise for this calculation, and use the vertically-integrated areal rate to extrapolate to the total gyre production rate.. We only had vertically integrated rates from the spring 2016 cruise, but the volumetric rate was three times higher in the summer of 2015. By using the spring rates, we are making a conservative estimate of TAG synthesis in the gyres. We agree with the reviewer that the calculation is basic, but there is a lack of data on TAG production rates for different seasons and for other locations within the gyre, which potentially have varying nutrient concentration. We are very clear in qualifying our calculation in the text...this is the first report of the phenomenon of daily TAG synthesis/production in the ocean, and it is incumbent on us to attempt to constrain its significance. Note: we constrain our calculations to the gyres, which relatively stable environments that share many attributes including low dissolved inorganic nitrogen, dominance of *Prochlorococcus* and a rich community of 2-20 μm eukaryotic phytoplankton.

(3) Perhaps minor, but frustrating for this reviewer...not including the primary production temporal and vertical variability in Fig 3 (as well as the POC, which may/may not have been measured on a diel basis) limits the interpretation for the reader – especially when this is a stated outcome (lines 77-79) of this paper. Even a supplementary figure would be welcome.

Although PP was not directly measured in water column profiles, we have now calculated a PP depth profile for March 2016. It was calculated by discretization of the below equation at the standard sampling depths of Station ALOHA:

$$P \text{ (mg C m}^{-3}\text{)} = 12,000 \times \text{PAR} \times [\text{chl } a] \times a^* \times \Phi_c \quad (\text{Eq. 1}),$$

where 12000 converts mol of carbon (C) to mg C, mean chlorophyll-specific absorption coefficient based on HPLC (a^* , 0.026) is as reported by Letelier et al. (for summer months and Φ_c , the quantum yield of photosynthesis [mol C mol quanta⁻¹] was parameterized from ¹⁴C-based photosynthesis-irradiance curves as per Li et al. (2015).

We included the calculation in the methods section and added the profile to original Fig. 3 (now Fig. 2).

We have now additionally included daily POC variability for both cruises in the Supplementary information (Supplementary Fig. 6).

Reviewer #5 (Remarks to the Author):

Title: Daily changes in phytoplankton lipidomes reveal mechanisms of energy storage in the open ocean
This review refers to the lipidomic analyses included in the work

In this manuscript, Becker and co-workers used a lipidomics to evaluate the effect of sunlight in phytoplankton, particularly in TAG.

The lipidomics approach used in this manuscript includes the initial step of lipid extraction, using a modified Bligh and Dyer protocol and an internal standard was added prior to the extraction to evaluate the lipid recovery. Reverse phase LC-MS was used to identify and quantify the lipidome of phytoplankton.

The absolute quantitation of each lipid molecular species was calculated from peak areas of the molecular ions in mass chromatograms using response factors from external standard curves and response factors. However no internal standards of each lipid class were added.

Recommendation for the revision to the manuscript:

Authors reported that they did absolute quantitation; however absolute quantitation requires the use of internal standards for each lipid class. There are several lipids standards in Avanti, either with odd chain fatty acids or deuterated lipids for all the phospholipid classes. In the case of betaines or glycolipids, you can choose one PL class with similar ionization efficiency. For example, you can use the PC standard, as internal standard for betaine class.

This is quite important particularly for reverse phase LC-MS lipid analysis, since in these cases there are co elution of ions from different classes, since elution of lipid species is quite dependent on the fatty acyl chains rather than the polar head group polarity. This contribute to ion suppression in the case of co elution of classes with different ionization capabilities, due to different ionization efficiencies for the lipid classes. This cannot be predicted with external calibration curves, that did not take into account the suppression effects in a lipid mixture.

If internal standard are no used you cannot talk about absolute quantitation but rather relative quantitation.

We have now conducted ion suppression experiments using deuterated internal standards as also suggested by Reviewer #2. For convenience, we have pasted the same text:

“Matrix effects in electrospray ionization of lipids tend to be most problematic in very rich samples (e.g. cultures) or samples with complex, often molecularly-uncharacterizable, co-extracting molecules (e.g. sediments). In our case, we are working with very dilute samples composed of >50% living biomass, composed of >90% molecularly characterizable molecules.

Nonetheless, as Reviewer #2 suggested, we have now conducted tests of matrix effects using a series deuterated standards. We now explicitly show that matrix effects did not affect our quantification, and we have included these results in the Supplementary Information of the revised manuscript.

We chose 30 samples from our archive (17 from the 2015 and 13 from the 2016 data set) and spiked with five deuterated standards including a TAG and a betaine lipid (DGTS) and compared the peak areas in the spiked samples with those from pure standard analysis (Fig. 1 below). On the x-axis are peak areas of the deuterated internal standards in these samples, and on the y-axis are the peaks areas of the deuterated standards alone in an identical amount of solvent. All 5 standard fall on the 1:1 line and thus, matrix effects had only a very little effect on lipid ionization and detection in our samples. This becomes even more apparent when the lipid concentrations in individual samples are corrected for ion suppression and are compared to the original concentrations (Fig. 2 below). It was decided, based on the results shown in Figures 1 and 2, that reanalyzing all the samples was not necessary, particularly considering the preciousness of our research funding and competing demands for instrument time associated with having undergraduates in the lab this summer.

Fig 1. Signal response comparisons for deuterated standards added to lipid extracts and deuterated standards measured alone. Cross plots of peak areas of deuterated standards versus deuterated standards added to lipid extracts from samples collected in spring 2016 (a) and summer 2015 (b). Error bars represent the standard deviation of averaged values with $n = 13$ for a and $n = 17$ for b. PE, phosphatidylethanolamine; DGTS, diacylglyceryltrimethyl-homoserine; PG, phosphatidylglycerol; TAG, triacylglycerol. The vertical error bars are smaller than the symbols.

Fig. 2. Effects of ion suppression on TAG concentrations from deuterated standard addition experiments. a, TAG concentration of triplicate samples from July 2015 before (closed symbols) and after (open symbols) correction for ion suppression. **b,** TAG concentration in samples from March 2016 before (black circles) and after (yellow circles) correction for ion suppression.”

Other suggestions

1) Information about the type of HPLC column used must be added.

We have included information about the HPLC column used and now describe the HPLC and MS settings in more detail.

2) Identification of the lipid species were only based on open source software? Did you confirm the identity of the lipid species by the analysis of raw data: LC-MS and MS/MS?

Yes, lipid identification are confirmed by analyzing the raw data. Representative lipids are identified by retention time as well as accurate molecular mass and isotope pattern matching of proposed sum formulas in full-scan mode and tandem MS (MS^2) fragment spectra.

3) Include in the supplementary file a list of all the lipid species identified and quantified, with observed mass, calculated mass and error (ppm). Provide information of fatty acyl composition, if possible.

We have included a table with the requested information in the supplementary file.

4) Include in supplementary file one MS and one MS/MS spectra representative of each class identified.

We have included a figure showing MS^2 spectra of representative lipids in the supplementary file.

5) Why did you not evaluate the variation of the lipid profile for the other lipid classes other than TAG?

Besides showing the molecular distributions of TAGs and ubiquinones, we have included figure showing the molecular distributions of DGCC for all samples analyzed in this study. Similar to the TAGs, the distribution is largely invariant suggesting a similar source for these lipids with time and depth.

A full description of the lipidome from these two cruise is forthcoming in a disciplinary journal.

Reviewers' Comments:

Reviewer #3:

Remarks to the Author:

The revised version is improved, but the discussion of the contribution of TAGs to oceanic primary production remains frustratingly imprecise. The authors' response to the previous comment on the distinction between net and gross primary production itself confuses the two ("do not [sic] contribute to gross production per liter"; "gross production at the scale of individual cells"). I will leave it to the Editor to decide whether such nuance is beyond the scope of a first paper on TAG synthesis for a journal like Nature Communications." There is one point in the text (l. 255-256) where the statement is made that, "the daily accumulation and depletion of TAGs in gyres could drive a carbon flux approaching 2.4 Pg C yr⁻¹, which corresponds to about 4.9% of global oceanic net primary production." Comparing the daytime accumulation of TAGs (12hr timescale) to global oceanic net primary production on an annual scale is totally misleading and inappropriate. The authors have declined (based on response to previous reviews) to comment quantitatively on the actual contribution of TAGs to net primary productivity on even daily timescales, so beyond any uncertainties about the geographic extrapolability of these data, this is a comparison of a short-term, gross flux with a long-term, net flux and is not meaningful.

Reviewer #5:

Remarks to the Author:

Authors replied to all queries .

Response to reviewer comments

Reviewer #3 (Remarks to the Author):

The revised version is improved, but the discussion of the contribution of TAGs to oceanic primary production remains frustratingly imprecise. The authors' response to the previous comment on the distinction between net and gross primary production itself confuses the two ("do no [sic] contribute to gross production per liter"; "gross production at the scale of individual cells"). I will leave it to the Editor to decide whether such "nuance is beyond the scope of a first paper on TAG synthesis for a journal like Nature Communications." There is one point in the text (l. 255-256) where the statement is made that, "the daily accumulation and depletion of TAGs in gyres could drive a carbon flux approaching 2.4 Pg C yr⁻¹, which corresponds to about 4.9% of global oceanic net primary production." Comparing the daytime accumulation of TAGs (12hr timescale) to global oceanic net primary production on an annual scale is totally misleading and inappropriate. The authors have declined (based on response to previous reviews) to comment quantitatively on the actual contribution of TAGs to net primary productivity on even daily timescales, so beyond any uncertainties about the geographic extrapolability of these data, this is a comparison of a short-term, gross flux with a long-term, net flux and is not meaningful.

To further clarify, if TAG synthesis mainly contributes to gross or net primary production, we have added the following text to the discussion (underlined and bold text represents new text):

"In addition to potentially impacting the timing of zooplankton grazing, the diel fluctuations in TAG content of eukaryotic phytoplankton are likely tightly coupled to the fate of organic matter within the NPSG. **If TAGs are respired entirely by phytoplankton, then their synthesis would not contribute to net primary production even though TAG production is a substantial fraction of gross production.** Using a simple model we estimate that around 70% of the TAGs produced during the day are consumed by eukaryotic phytoplankton themselves, while 30% is consumed by mortality processes (see Supplementary Methods)." (lines 266-272)

We understand the reviewer's concerns regarding our calculations of the contribution of TAG synthesis to global oceanic net primary production and deleted this statement in our revised version.

Reviewer #5 (Remarks to the Author):

Authors replied to all queries .

We are pleased that the reviewer is satisfied with our changes and corrections.